# Resurrection of a Viral Internal Ribosome Entry Site from a 700 Year Old Ancient Northwest Territories Cripavirus

**DOI:** 10.3390/v13030493

**Published:** 2021-03-17

**Authors:** Xinying Wang, Marli Vlok, Stephane Flibotte, Eric Jan

**Affiliations:** 1Department of Biochemistry and Molecular Biology, University of British Columbia, Vancouver, BC V6T 1Z3, Canada; ubcamy69@student.ubc.ca (X.W.); marlivlok@alumni.ubc.ca (M.V.); 2UBC/LSI Bioinformatics Facility, University of British Columbia, Vancouver, BC V6T 1Z3, Canada; stephane.flibotte@ubc.ca

**Keywords:** dicistrovirus, RNA, internal ribosome entry site, translation, ribosome, infectious clone

## Abstract

The dicistrovirus intergenic region internal ribosome entry site (IGR IRES) uses an unprecedented, streamlined mechanism whereby the IRES adopts a triple-pseudoknot (PK) structure to directly bind to the conserved core of the ribosome and drive translation from a non-AUG codon. The origin of this IRES mechanism is not known. Previously, a partial fragment of a divergent dicistrovirus RNA genome, named ancient Northwest territories cripavirus (aNCV), was extracted from 700-year-old caribou feces trapped in a subarctic ice patch. The aNCV IGR sequence adopts a secondary structure similar to contemporary IGR IRES structures, however, there are subtle differences including 105 nucleotides upstream of the IRES of unknown function. Using filter binding assays, we showed that the aNCV IRES could bind to purified ribosomes, and toeprinting analysis pinpointed the start site at a GCU alanine codon adjacent to PKI. Using a bicistronic reporter RNA, the aNCV IGR can direct translation in vitro in a PKI-dependent manner. Lastly, a chimeric infectious clone swapping in the aNCV IRES supported translation and virus infection. The characterization and resurrection of a functional IGR IRES from a divergent 700-year-old virus provides a historical framework for the importance of this viral translational mechanism.

## 1. Introduction

Due to their limited genomes, viruses do not encode ribosomes nor a full complement of translation factors and as a result, they must hijack the host translation machinery to drive viral protein synthesis [1]. Most eukaryotic mRNAs utilize a core of eukaryotic initiation factor (eIFs) to mediate the 5′ cap-dependent scanning mechanism that involves recruitment of the 40S ribosomal subunit and initiator Met-tRNAi, scanning by the pre-initiation complex along the 5′ untranslated region (UTR) to locate the appropriate AUG start codon and 60S ribosomal subunit joining to form an 80S elongation competent ribosome. Some virus infections lead to shutdown of overall host cap-dependent translation, either as an antiviral response or as a viral strategy [2]. However, viruses have evolved mechanisms to bypass the translation block and compete for the recruitment of the ribosome to facilitate viral translation.

One such mechanism is through viral *cis*-acting elements, called internal ribosome entry sites (IRESs), which enable mRNAs to recruit ribosome internally and initiate translation [1]. Among different types of IRESs, the dicistrovirus intergenic (IGR) IRES uses the most autonomous and streamlined mechanism by directly recruiting the ribosome and starting translation at a non-AUG codon [3,4]. Dicistroviruses are positive sense, single-stranded RNA viruses that primarily infect arthropods [5]. The approximately 9 kb genome of dicistroviruses contains two open reading frames (ORFs) encoding viral non-structural and structural proteins, both of which are driven by distinct IRESs. Translation of the first ORF is directed by an IRES located at the 5′ UTR. The IGR IRES controls translation of the downstream ORF encoding the structural proteins. 

The key to the IGR IRES mechanism is within its core secondary and tertiary structures. Typically, 150–200 nucleotides in length, the IGR IRES contains three main pseudoknots (PKI-III) that direct distinct functions; PKII and PKIII form a core domain that is responsible for recruiting 40S and 60S ribosomal subunits, while PKI contains a tRNA-like anticodon:codon structure that occupies the ribosomal A-site [1,6,7]. Structural and biochemical analyses have revealed key contacts between the IGR IRES and the ribosome that drive factorless translation. In the initial assembly of ribosomes on the IGR IRES, stem-loops SLIV and SLV interact with uS7 and eS25 of the 40S subunit, and loop L1.1 interacts with the L1 stalk of the 60S subunit [6]. Mutations of these terminal loop regions disrupt 40S and/or 60S recruitment. After ribosome assembly, the IRES:ribosome complex undergoes a translocation event whereby the PKI domain, which initially occupies the ribosomal A site, translocates to the P site and allows delivery of an aminoacyl-tRNA to the empty ribosomal A site. Both of these events are coordinated by elongation factors eEF1A and eEF2 and are called pseudo-translocation as this does not involve peptide formation [8,9,10]. The IRES undergoes a second pseudo-translocation event leading to movement of the aminoacyl-tRNA to the P site and delivery of the next aminoacyl-tRNA to the A site, where at this point peptide formation can occur. Cryo-EM analysis capturing the IRES translocating through the ribosome has revealed that the IRES mediates interactions within all three tRNA binding sites of the ribosome [10,11,12]. Remarkably, the IRES undergoes a dynamic conformational change after two translocation cycles whereby the PKI domain in the ribosomal E site flips from an anticodon-codon mimic to a tRNA acceptor arm mimic on the ribosome [7]. IRES binding within and manipulation of the conserved core of the ribosome explain how this IRES can function across species including yeast, plant, insect and mammalian systems [13,14,15]. The IRES can also recruit and direct translation in bacteria, though the mechanism appears to be distinct [13]. 

Members of this family include cricket paralysis virus (CrPV) and Drosophila C virus, which have been studied extensively and used as models for understanding insect innate immunity and fundamental viral translation mechanisms including the IGR IRES [5]. Other members include the Taura syndrome virus, which has led to outbreaks of disease in penaeid shrimp, plautia stali intestine virus, which infects aphids, and the honeybee viruses, Israeli acute bee paralysis virus, acute bee virus and Kashmir bee virus, which have been associated with bee disease [5,16]. The mechanism of IGR IRESs of CrPV, TSV and IAPV have been studied at the biochemical and structural level [7,17,18].

The origin of the IGR IRES mechanism is not known, but analysis of present-day IRESs may provide insights. Currently, the IGR IRESs studied to date can be divided into two main types; Type I and II IGR IRESs are exemplified by the CrPV and IAPV IGR IRES, respectively, with the main differences in the L1.1 nucleotides, and that the Type II IGR IRESs have an extra stem-loop within the PKI domain [19]. Previous studies have shown that the PKI domains of Type I and II can be functionally interchanged and that the IRESs are modular, thus suggesting that the IRESs may have evolved through recombination of modular domains [20,21]. Recent metagenomic approaches have identified an increasing diversity of dicistro-like viral genomes and IGR IRESs that may provide hints into the origins of the IRES [22,23]. Alternatively, identifying ancient RNA viral genomes may provide historical context for the evolution of viral strategies; however, this is challenging given the relatively labile nature of RNA. However, some preserved RNA viral genomes have been discovered. The pioneering benchmark in identifying ancient RNA viruses is the recovery of the influenza virus genome from 1918–1919 [24,25]. Furthermore, a compete genome of ancient Barley Stripe Mosaic Virus was identified from barley grain dated (~700 years) and a 1000 year old RNA virus related to plant chryso-viruses was isolated from old maize samples with a nearly complete genome [26,27]. Recently, Ng et al. recovered two novel viruses from 700-year-old caribou feces trapped in a subarctic ice patch [28], one of which is a fragment of a divergent dicistrovirus RNA genome, named ancient Northwest territories cripavirus (aNCV). The partial aNCV sequence that was recovered included the IGR domain, thus providing an opportunity to characterize and compare the aNCV IGR to contemporary dicistrovirus IGR IRESs. In this study, we examine the molecular and biochemical properties of the aNCV IGR and demonstrate that the aNCV IGR directly assembles ribosomes and can direct internal ribosome entry both in vitro and in vivo. We show that the intact aNCV IGR including the IRES region and 105 nucleotides upstream of the IRES can support viral translation and infection in a heterologous dicistrovirus clone, thus highlighting the significance of this translational mechanism in an ancient virus.

## 2. Materials and Methods

### 2.1. Cell Culture

*Drosophila Schneider* line 2 (S2) cells were maintained and passaged at 25 °C in Shields and Sang M3 insect medium (Sigma-Aldrich, Oakville, ON, Canada) supplemented with 10% fetal bovine serum.

### 2.2. Virus Infection

S2 cells were infected at the desired multiplicity of infection in minimal phosphate-buffered saline (PBS) at 25 °C. After 30 min absorption, complete medium was added and harvested at the desired time point. Virus titres were monitored as described [29] using immunofluorescence (anti-VP2).

### 2.3. DNA Constructs

The aNCV IGR sequence (Accession KJ938718.1) was synthesized (Twist Biosciences, San Francisco, CA, USA) and cloned into pEJ4 containing EcoRI and NcoI sites upstream of the firefly luciferase (FLuc) open reading frame. For the bicistronic reporter construct, PCR-amplified full-length aNCV IGR or truncated aNCV IGR (∆1-99) was ligated into the standard bicistronic reporter construct (pEJ253) using EcoRI and NdeI sites.

### 2.4. CrPV/aNCV Chimeric Infectious Clone Constructs

The chimeric CrPV-aNCV infectious clone was derived from the full-length CrPV infectious clone (pCrPV-3; Accession KP974707) [30] using Gibson assembly (New England BioLabs, Whitby, ON, Canada), per the manufacturer’s instructions. All constructs were verified by sequencing. 

### 2.5. In Vitro Transcription and Translation

Mono-cistronic and bicistronic reporters were linearized with NcoI and BamHI, respectively. pCrPV-3 and chimeric CrPV-aNCV clones were linearized with Eco53kI. RNAs were in vitro transcribed using a bacteriophage T7 RNA polymerase reaction and RNA was purified using a RNeasy Kit (Qiagen, Toronto, ON, Canada). Radiolabeled RNAs were bulk labelled by incorporating α- [^32^P] UTP (3000 Ci/mmol). The integrity and purity of RNAs were confirmed by agarose gel analysis. Uncapped bicistronic RNAs were first pre-folded by heating at 65 °C for 3 min, followed by the addition of 1xbuffer E (final concentration: 20 mM Tris pH 7.5, 100 mM KCl, 2.5 mM MgOAc, 0.25 mM Spermidine, and 2 mM DTT) and slowly cooled at room temperature for 10 min. The pre-folded RNAs (20–40 ng/μL) were incubated in RRL containing 8 U Ribolock inhibitor (Thermo Fisher Scientific, Mississauga, ON, Canada), 20 μM amino acid mix minus methionine, 0.3 μL [^35^S]-methionine/cysteine (PerkinElmer, >1000 Ci/mmol), and 75 mM KOAc pH 7.5 at 30 °C for 1 hr. For the infectious clones, 2 μg RNA was incubated at 30 °C for 2 hr in *Spodoptera frugiperda* (Sf21) extract (Promega, Madison, WI, USA) in the presence of [^35^S]-methionine/cysteine (PerkinElmer) and an additional 40 mM KOAc and 0.5 mM MgCl_2_. The translated proteins were resolved using SDS-PAGE and analyzed by phosphor-imager analysis (Typhoon, GE life sciences, Mississauga, ON, Canada).

### 2.6. Purification of the 40S and 60S Subunits

Ribosomal subunits were purified from HeLa cell pellets (Cell Culture Company, Minneapolis, MN, USA) as described [31]. In brief, HeLa cells were lysed in a lysis buffer (15 mM Tris–HCl (pH 7.5), 300 mM NaCl, 6 mM MgCl_2_, 1% (*v/v*) Triton X-100, 1 mg/mL heparin). Debris was removed by centrifuging at 23,000 × g and the supernatant was layered on a 30% (*w/w*) cushion of sucrose in 0.5 M KCl and centrifuged at 100,000 × g to pellet crude ribosomes. Ribosomes were gently resuspended in buffer B (20 mM Tris–HCl (pH 7.5), 6 mM magnesium acetate, 150 mM KCl, 6.8% (*w/v*) sucrose, 1 mM DTT) at 4 ^o^ C, treated with puromycin (final 2.3 mM) to release ribosomes from mRNA, and KCl (final 500 mM) was added to wash and separate 80S ribosomes into 40S and 60S. The dissociated ribosomes were then separated on a 10%–30% (*w/w*) sucrose gradient. The 40S and 60S peaks were detected by measuring the absorbance at 260 nm. Corresponding fractions were pooled and concentrated using Amicon Ultra spin concentrators (Millipore Sigma, Oakville, ON, Canada) in buffer C (20 mM Tris–HCl (pH 7.5), 0.2 mM EDTA, 10 mM KCl, 1 mM MgCl_2_, 6.8% sucrose). The concentration of 40S and 60S subunits was determined by spectrophotometry, using the conversions 1 A260 nm = 50 nM for 40S subunits, and 1 A260 nm = 25 nM for 60S subunits. 

### 2.7. Filter-Binding Assays

RNAs (final 0.5 nM) were preheated at 65 °C for 3 min, followed by the addition of 1xbuffer E and slowed cooling in a water bath preheated to 60 °C for 20 min. The pre-folded RNAs, with 50 ng/μL of non-competitor RNA, were incubated with an increasing amount of 40S subunits from 0.1 nM to 100 nM, and a 1.5-fold excess of 60S subunits for 20 min at room temperature. Non-competitor RNAs were in vitro transcribed from the pcDNA3 vector 880–948 nucleotides. Reactions were then loaded onto a Bio-Dot filtration apparatus (Bio-Rad, Hercules, CA, USA) including a double membrane of nitrocellulose and nylon pre-washed with buffer E. Membranes were then washed three times with buffer E, dried, and the radioactivity was imaged and quantified by phosphor-imager analysis. The dissociation constant is determined by the formula,
(1)$$\frac{\left[AB\right]}{{\left[A\right]}_{total}}=f_{max}\left(\frac{\left[B\right]}{\left[B\right]+K_{D}}\right)$$
where [*A*] is the concentration of RNAs, [*B*] is the concentration of ribosomes, [*AB*] is the concentration of RNAs bound to the ribosomes, fmax is the saturation point, and *K_D_* is the dissociation constant.

Bulk-labeled RNAs (final 0.5 nM), IRES competitors, and non-competitor RNAs (50 ng/μL) were pre-folded in buffer E. Unlabeled IRES competitors were added in increasing concentrations from 2 nM to 250 nM. RNAs were then incubated with 6 nM 40S and 9 nM 60S subunits at room temperature for 20 min. Reactions were then loaded onto the Bio-Dot filtration apparatus, and data were fitted to the Linn-Riggs equation that describes competitive ligand binding to the target: (2)$$\theta =\;\frac{\left[S\right]\left(1-\theta \right)}{K_{D}\left\{\left(1+\left[C\right]\right)K_{C}\right\}+\left[R\right]\left(1-\theta \right)}$$
where [*S*] is 80S ribosome concentration, [*R*] is radiolabelled RNA concentration, *θ* is fraction of radiolabelled RNAs bound to ribosomes, [*C*] is competitor RNA concentration, *K_D_* and *K_C_* are dissociation constants of labelled RNAs and competitor RNAs, respectively.

### 2.8. Ribosome Protection Assay

[^32^P]-labeled RNAs (final 0.1 μM) were pre-folded in buffer E and incubated with 0.6 μM 40S and 0.9 μM 60S at room temperature for 20 min. 1 μL of 1 U/μL of RNase I (Thermo Fisher Scientific, Mississauga, ON, Canada) was added to the mixture and incubated at 20 °C for 1 hr. RNAs without RNase I treatment were incubated at 20 °C with the same time length. RNAs from mixtures with or without RNase I treatment were TRIZOL-extracted and loaded onto 6% (*w/v*) polyacrylamide/8M urea gels to separate them. RNA Ladders (Thermo Fisher Scientific) were synthesized per manufacturer’s instruction. Gels were dried and imaged by phosphor-imager analysis. 

### 2.9. Toeprinting/Primer Extension Analysis

Toeprinting analysis of ribosomal complexes in RRL was performed as previously described [3]. 0.4 µg of bicistronic WT or mutant CrPV IGR IRES RNAs and WT or mutant aNCV IGR IRES RNAs were annealed to primer 5′-GTAAAAGCAATTGTTCCAGGAACCAG-3′ and primer 5′-GTTAGCAGACTTCCTCTGCCCTCTC-3′, respectively in 40 mM Tris (pH 7.5) and 0.2 mM EDTA by slow cooling from 65 °C to 30 °C. Annealed RNAs were added to RRL pre-incubated with 0.68 mg/mL cycloheximide and containing 20 µM amino acid mix, 8 units of Ribolock (Thermo Fisher Scientific, Mississauga, ON, Canada ) and 154 nM final concentration of potassium acetate (pH 7.5). The reaction was incubated at 30 °C for 20 min. Toeprinting analysis using purified ribosomes was performed as follows: 75 ng of RNAs were annealed to primers in 40 mM Tris·Cl, pH 7.5, and 0.2 mM EDTA by slow cooling from 65 °C to 35 °C. Annealed RNAs were incubated in buffer E (containing 100 mM KCl) containing 40S (final concentration 100 nM); 60S (final concentration 150 nM); ribo-lock (0.02 U/μL) at 30 °C for 20 min. Following incubation, ribosome positioning was determined by primer extension/reverse transcription using AMV reverse transcriptase (1 U/μL) (Promega, Madison, WI, USA) in the presence of 125 µM of each of dTTP, dGTP, dCTP, 25 µM dATP, 0.5 µL of α- [^32^P] dATP (3.33 µM, 3000 Ci/mmol), 8 mM MgOAc, in the final reaction volume. The reverse transcription reaction was incubated at 30 °C for 1 h, after which it was quenched by the addition of STOP solution (0.45 M NH4OAc, 0.1% SDS, 1 mM EDTA). Following the reverse transcription reaction, the samples were extracted by phenol/chloroform (twice), chloroform alone (once), and ethanol precipitated. The cDNA was analyzed under denaturing conditions on 6% (*w/v*) polyacrylamide/8M urea gels, which were dried and subjected to phosphor-imager analysis.

### 2.10. RNA Transfection

3 μg of in vitro-transcribed RNA derived from the pCrPV-3 or chimeric clones was transfected into 3 × 10^6^ S2 cells using lipofectamine 2000 reagent (Thermo Fisher Scientific, Mississauga, ON, Canada) per the manufacturer’s instructions. 

### 2.11. RT-PCR and Sequence Confirmation

Viruses were passaged three times after transfection of viral RNAs in S2 cells. Total RNA was isolated from S2 cells using TRIzol reagent. RT was performed using 1 μg of RNA using LunaScript™ RT SuperMix Kit (New England BioLabs, Ipswich, MA, USA) per manufacturer’s instructions. For reverse transcription of the negative-sense CrPV viral RNA to detect replication, tagged primer (5′-CTATGGATCCATGGGAGAAGATCAGCAAAT-3′; tag is underlined) was used. Primer (5′-CTATGGATCCATGGGAGAAG-3′) and primer (5′-GTGGCTGAAATACTATCTCTGG-3′) were used for PCR amplification of the negative-sense strand of the CrPV genome. Rps6 was amplified using primers (5′-CGATATCCTCGGTGACGAGT-3′) and (5′-CCCTTCTTCAAGACGACCAG-3′). 

### 2.12. Western Blotting

Cells were washed once using 1 × PBS and harvested in lysis buffer (20 mM HEPES, 150 mM sodium chloride, 1% Triton X-100, 10% glycerol, 1 mM EDTA, 10 mM tetra-pyrophosphate, 100 mM sodium fluoride, 17.5 mM β-glycerophosphate, protease inhibitor cocktail (Sigma-Aldrich, St. Louis, MO, USA). Equal amounts of protein lysate were resolved by SDS-PAGE and subsequently transferred onto polyvinylidene difluoride Immobilon-FL membrane (Millipore Sigma, Burlington, MA, USA). Following transfer, the membrane was blocked for 30 min at room temperature in 5% skim milk in Tris-buffered saline containing 0.1% Tween 20 (TBST) and incubated for 1 h with rabbit polyclonal antibody raised against CrPV 1A (1:1000; Genscript, Piscataway, NJ, USA) or CrPV VP2 (1:5000; Genscript). Membranes were washed three times with TBST and subsequently incubated with IRDye 800CW goat anti-rabbit IgG (1:20,000; Li-Cor Biosciences, Lincoln, NE, USA) for 1 h at room temperature.

### 2.13. Phylogenetic Analysis

Nine representative untrimmed Type I IRES were aligned with muscle [32] and their consensus secondary structure was obtained using the RNAalifold tool from the ViennaRNA package [33]. Selected IRES were then aligned to a covariance model built with the Infernal software [34] taking into account sequence and secondary structure. Those alignments were subsequently used as input to build phylogenetic trees. Maximum-likelihood trees were constructed with PhyML 3.0 [35] and the HKY85 +G nucleotide model was selected using Smart Model Selection [36]. Branch support was evaluated with the Shimodaira-Hasegawa approximate-likelihood ratio test and resultant trees were edited in iTOL v3 [37].

## 3. Results

### 3.1. aNCV IGR IRES Adopts a Triple Pseudoknot Structure

Using computational and compensatory base-paring analysis, a secondary structure model of the aNCV IGR was predicted (Figure 1A). The aNCV IGR is predicted to adopt an overall similar structure to modern dicistrovirus IGR IRESs, all possessing the main PKI, PKII and PKIII structures and is classified as a Type I IGR IRES. However, there were notable differences. An alignment analysis of the aNCV IGR with classic Type I and II IGR IRESs showed that the aNCV IGR structure contains a chimera mix of Type I and II features, especially at key domains that direct distinct steps of IGR IRES translation (Figure 1B). Specifically, the aNCV IGR shares sequence similarities to Type I IGR IRESs within Loop 1.1B, Loop 3, and to Type II at Loop 1.1A. Furthermore, whereas all Type I and II IGR IRESs contain an invariant AUUU within the loop of SLIV, the aNCV contains an AUUA loop sequence. Strikingly, the aNCV IGR IRES also includes an extra 105 nucleotides between the stop codon of ORF1 and the start of the predicted IGR IRES structure. Several smaller stem-loops (SLVI, SLVII and SLVIII) are predicted within this 105-nucleotide region. In summary, the aNCV IGR adopts an overall secondary structure that is similar to other known dicistrovirus IGR IRESs but is a chimera of Type I and II IGR IRESs at key loop domains and has an extended upstream region.

Based on a maximum-likelihood phylogeny constructed from a nucleotide alignment informed by RNA structure (Figure 1C), the aNCV IRES is most closely related to that of the Homalodisca coagulata virus-1 type I IRES. The branch lengths for both ANCV and Homalodisca coagulata virus-1 are long, suggesting more genetic change since their last common ancestor, compared to CrPV and DCV which are more closely related.

### 3.2. aNCV IGR IRES Binds Tightly to Human 80S Ribosomes

Given the chimeric makeup of the aNCV IGR, we next examined whether the aNCV IGR possesses properties similar to dicistrovirus IGR IRESs. The first property tested was its ability to bind to purified ribosomes in vitro. The IGR IRESs directly bind to purified ribosomes with high affinity [31,38]. Using an established filter-binding assay, we measured 80S binding to the aNCV IGR and the CrPV IGR IRES by incubating radiolabeled IGR with increasing amounts of purified salt-washed human 40S and 60S. The fraction of ribosome: IGR complexes was then resolved by monitoring the amount of radioactivity on the nitro-cellular and nylon membrane. As shown previously, the wild type but not the mutant (∆PKI/II/III) CrPV IGR IRES bound to 80S ribosomes with high affinity (apparent K_D_ 0.4 nM) [31,38]. The mutant CrPV ∆PKI/II/III IRES contains mutations that disrupts all three PK base-pairings (Figure 2A). Similarly, the wild type aNCV IGR bound to purified ribosomes with a similar affinity as the CrPV IGR IRES (K_D_ 0.7 nM). To confirm the binding specificity of ribosome:IGR complex interactions, we performed competition assays by addition of excess unlabeled IGR RNAs to the reaction prior to incubating with ribosomes. As expected, adding increasing amounts of unlabeled wild type but not mutant CrPV IGR decreased the levels of radiolabeled CrPV IGR bound to 80S ribosomes (Figure 2B, left). Similarly, adding increasing amounts of unlabeled aNCV IGR also decreased the fraction of CrPV IGR IRES-ribosome complex formation. In the reverse experiment, excess of unlabeled wild-type CrPV and aNCV IGRs but not mutant CrPV IGR also competed for ribosomes from radiolabeled aNCV (Figure 2B, right). The apparent K_D_ measurements of the aNCV-ribosome complex binding in the competition assay were similar to the direct filter binding assay (K_D_ 0.8 nM). Taken together, aNCV IGR RNA binds to purified ribosomes with high affinity and is likely to occupy the same sites on the ribosome as the CrPV IGR IRES.

### 3.3. RNase Protection Analysis of aNCV IGR-Ribosome Complexes

We next examined whether aNCV IGR RNA binds within the inter-subunit space of the ribosome. We previously developed a novel ribosome protection assay, whereby localization of the RNA in the inter-subunit space or solvent side of the ribosome can be inferred by its susceptibility to RNase I-mediated degradation (data not shown). Radiolabeled CrPV IGR IRES in complex with purified ribosomes were incubated with RNase I and then the RNA was isolated and resolved on a urea-PAGE gel. RNase I treatment of radiolabeled CrPV IGR IRES prebound to the ribosome resulted in faster migrating fragments, indicative of sequences that were protected by the ribosome from degradation (Figure 3). Specifically, compared to the full-length CrPV IGR IRES (188 nucleotides), RNase I treatment led to ribosome-protected fragments ranging from 50–160 nucleotides. Sequencing of these IRES fragments and mapping back to the IRES revealed a signature core domain consisting of PKII and PKIII structures that were primarily protected by the ribosome from RNase I (unpublished work). Importantly, RNase I treatment of the mutant ΔPKI/II/III CrPV IGR IRES (TM) incubated with ribosomes did not lead to protected fragments, indicating that the concentration of RNase I is sufficient to degrade the unprotected RNA. Similar to that observed with the CrPV IGR IRES, RNase I-treatment of aNCV IGR-ribosome complexes resulted in faster migrating fragments (Figure 3). Compared to the full-length aNCV IGR IRES (291 nucleotides), several protected fragments ranging from 70 to 200 nucleotides were detected. These results indicate that the aNCV IGR binds to the inter-subunit core of the ribosome. 

### 3.4. Determination of the aNCV IGR IRES Initiation Site

We next determined the start site of the aNCV IGR, which is predicted to be at a GCU codon and adjacent to the PKI domain (Figure 1). To investigate this, we monitored initiating ribosomes assembled on the aNCV IGR in rabbit reticulocyte lysates (RRL) by toeprinting, an established primer extension assay [39]. Briefly, when reverse transcriptase encounters the ribosome assembled on the IRES, a truncated cDNA product is generated, which can be detected on a urea-PAGE gel. As shown previously, ribosomes assembled on the wild-type but not mutant CrPV IGR IRES in RRL in the presence of cycloheximide resulted in toeprints at CC6232-3, which is +20-21 nucleotides from the PKI domain CCU triplet, given that the first C is +1 (Figure 4B) [31]. This result showed that the ribosome can translocate on the IGR IRES two cycles in the presence of cycloheximide [3,8,9]. Ribosomes assembled on the aNCV IGR in the presence of cycloheximide resulted in a prominent toeprint at U308, which is +20 nucleotides from the AUC codon, given that A is +1, which is consistent with ribosomes having translocated two cycles (Figure 4B). To confirm that this toeprint is representative of translocated ribosomes on the aNCV IGR, mutations within the PKI domain were generated that disrupted PKI base-pairing (mPKI #1 and #2) or compensatory (comp) mutations that restored PKI base-pairing (Figure 4B). Toeprint U308 was only detected when the PKI domain is intact, thus supporting the conclusion that the initiating codon is the GCU alanine adjacent to the PKI of aNCV IGR. 

To further confirm direct ribosome binding on the IRES, we performed toeprinting analysis of purified ribosomes assembled on the aNCV IGR IRES. As shown previously, purified ribosomes assembled on the wild-type but not mutant CrPV IGR IRES resulted in toeprints at CA6226-7, which is +13-14 nucleotides from the PKI domain CCU triplet, given that the first C is +1 (Figure 4C). Purified ribosomes assembled on the aNCV IGR resulted in a toeprint at C302, which is +13 nucleotides from the AUC codon, given that A is +1 (Figure 4C), thus supporting the conclusion that ribosomes initially assembled on the aNCV IGR contain the AUC codon in the ribosomal A site and subsequently, after the first pseudo-translocation step, the adjacent start site GCU codon occupies the A site where IRES initiation starts from.

### 3.5. aNCV IGR IRES Directs Translation in Vitro

To determine whether the aNCV IGR has IRES activity, we inserted the aNCV IGR in a bicistronic reporter RNA construct and assessed translation in RRL (Figure 5A,B). In vitro transcribed RNA was incubated in RRL in the presence of [^35^S]-methionine/cysteine to monitor protein expression level. Wild type but not mutant (mPKI) aNCV IGR IRES was active in RRL in vitro (Figure 5A lane 3). Importantly, mutating the aNCV PKI region in the reporter abolished FLuc activity, indicating that IRES activity is compromised (Figure 5A, lanes 4 and 5). In contrast, compensatory mutations that restored base-pairing rescued aNCV IGR IRES activity to ~50% of wild type (Figure 5A, lane 6). Unlike some dicistrovirus IGR IRESs that can direct +1 frame translation [17,40], the aNCV IGR IRES does not do this in RRL (data not shown). In summary, the aNCV IGR is a bona fide IRES.

To investigate whether the upstream region of the aNCV IRES is important for IRES activity, we monitored translation of wild type and a mutant aNCV IRES, where the upstream region has been deleted (Δ1-99). In RRL, both wild type and mutant (Δ1-99) aNCV can direct IRES activity to similar levels (Figure 5B), indicating that this upstream region does not contribute to IRES activity in vitro. 

### 3.6. Chimeric CrPV Clone Containing the aNCV IGR Is Infectious 

Having demonstrated that the aNCV IGR can bind to ribosomes directly and drive IRES activity from a non-AUG codon, we next examined whether the aNCV can support infection in a more physiological system. Because only a partial sequence of the aNCV genome was recovered in the 700-year-old caribou feces [28], we used a heterologous approach by generating chimeric dicistrovirus CrPV clones by replacing IGR IRES with either the full-length aNCV (CrPV-aNCV) or the mutant aNCV (Δ1-99), where the upstream 99 nucleotides is deleted [29,30] (Figure 6A). We first addressed whether the aNCV IGR can support translation in the infectious clone. In Sf21 extracts, incubation of in vitro transcribed chimeric CrPV-aNCV led to translation of non-structural and structural proteins that are similar to that of the wild-type CrPV infectious clone (Figure 6B). Of note, the chimeric CrPV-aNCV containing either the full-length or the mutant aNCV (Δ1-99) led to similar expression of viral proteins in vitro, consistent with previous data showing that the upstream 99 nucleotides do not affect aNCV IRES activity. Compared to the CrPV clone RNA, translation of structural proteins VP1 and VP2/3 was compromised in reactions containing the full-length and the mutant chimeric CrPV-aNCV RNA, indicating a defect in aNCV IGR IRES translation in vitro under these conditions (Figure 6B). The mutant CrPV containing a stop codon within ORF1 only resulted in expression of the unprocessed ORF2 polyprotein [30].

To determine whether the CrPV-aNCV clone is infectious, we transfected in vitro transcribed RNA into S2 cells and monitored viral protein expression by immunoblotting. Transfection of the wild-type CrPV and CrPV-aNCV containing the full-length aNCV IGR RNA resulted in expression of the CrPV non-structural protein 1A and the structural protein VP2 at 120 h post-transfection (h.p.i) (Figure 6C), which from previous experience is indicative of productive infection [30]. By contrast, viral proteins were not detected in cells transfected with the CrPV-aNCV (Δ1-99) clone. To confirm these results, we monitored viral replication by RT-PCR analysis. Mirroring the viral protein expression, CrPV RNA was detected by RT-PCR after transfection of the CrPV-aNCV and CrPV RNA but not the CrPV-aNCV (Δ1-99) RNA (Figure 6D). Note that the CrPV RNA was detected at 72 h h.p.t. whereas CrPV-aNCV was not detected until 144 h.p.t., suggesting that the replication of the chimeric virus is delayed.

We attempted to propagate the CrPV-aNCV virus from transfected cells by reinfecting and passaging in naïve S2 cells. Despite multiple attempts, the CrPV-aNCV (Δ1-99) did not lead to productive virus as measured by viral titres. However, the CrPV-aNCV yielded productive virus (1.15 X 10^10 FFU/mL). We sequence verified that the aNCV IGR was intact in viral RNA isolated from the propagated CrPV-aNCV (data not shown). To further validate virus production, we infected S2 cells with CrPV-aNCV and monitored viral expression by immunoblotting. Similarly to that observed with CrPV infection, the CrPV 1A protein produced from CrPV-aNCV was detected at 2-10 h.p.i., albeit VP2 expression was reproducibly detected until 4 h.p.i., thus likely reflecting the decreased IRES activity of aNCV compared to CrPV (Figure 6E). In summary, we have demonstrated that the full-length aNCV IGR can support virus infection in a heterologous infectious clone.

## 4. Discussion

Tracking back and identifying ancient RNA viruses may provide insights into the evolution and origin of present-day viruses including the mechanisms that permit virus translation, replication and host virus interactions. In this study, we have examined the IGR of an ancient dicistrovirus that was discovered from 700-year-old ice-preserved caribou feces. To our knowledge, the aNCV genome has not be identified in RNA metagenomic analysis; thus, this study is the first to characterize the oldest IRES to date and provides insights into the origins of this type of IRES. Using molecular and biochemical approaches, we demonstrate that the aNCV IGR possesses IRES activity using a mechanism that is similar to that found in present day Type IV dicistrovirus IRESs. The aNCV IGR can direct ribosome assembly directly and initiates translation from a non-AUG codon. The aNCV IGR IRES can support virus infection in a heterologous infectious clone; thus, a functional aNCV IGR IRES has been resurrected from an ancient RNA virus. 

The secondary structure model of the core aNCV IGR resembles classic Type I IGR IRESs containing all three PKs. However, there are subtle differences, including the L1.1 bulge region, responsible for 60S recruitment, which is comprised of a mixture of Type I and II IRESs elements. Additionally, aNCV IGR contains 105 nucleotides upstream of the core IGR IRES, within which several stem-loops are predicted. We showed that this upstream sequence is not required for aNCV IGR IRES translation (Figure 5 and Figure 6) but is required for virus infection using the CrPV-aNCV. RT-PCR analysis of the viral RNA suggests that the upstream region has a role in replication; however, it may have other roles in the viral life cycle such as RNA stability or replication. Of note, a few dicistrovirus IGRs also contain sequences beyond the core IRES structure. For example, the Rhopalosiphum padi virus (RhPV) contains an IGR that is over 500 nucleotides. Similar to the aNCV IRES, the extra sequences within the RhPV IGR do not affect the Type I RhPV IGR IRES [41]. It will be interesting to investigate in more detail how these extra IGR sequences play a role in the viral life cycle. 

To date, there are increasing numbers of dicistrovirus-like genomes identified via metagenomic studies [22,23]. From limited analysis, it is clear that the IGR IRES structures can be distinct (i.e., Type I and II) and can have novel functions (i.e., +1 reading frame selection). For example, the Halastavi árva virus IRES uses a unique but similar streamlined mechanism as the CrPV IRES, where it bypasses the first pseudo-trans-location event to direct translation initiation [42]. Moreover, the IGR IRES can direct translation in multiple species [13,14,15], suggesting that the IGR IRES mechanism may be a molecular fossil of an RNA-based translational control strategy that existed in ancient viruses. IGR IRESs are likely to have evolved through recombination of independent functional domains; the PKI domains can be functionally swapped between Type I and II IRESs [20,21]. Furthermore, IRES elements in unrelated viruses appear to have disseminated between these viruses via horizontal gene transfer [43], further supporting the idea that dicistrovirus IGR IRESs may have originated via recombination events. 

Given that the aNCV IGR IRES structure resembles an IGR IRES found in contemporary dicistrovirus genomes, it was not surprising that it functions by a similar mechanism. Relatively speaking, the IGR IRES from a 700-year-old RNA viral genome is far from the presumed primordial IRES. However, this study provides proof-in-concept that this viral RNA translation mechanism can be resurrected and investigated to provide context in the evolution of viral mechanisms. The challenge in identifying more ancient RNA viral genomes and mechanisms has been and will always be in capturing intact RNA viral genomes as RNA, compared to DNA virus counterparts, are unstable and apt to degrade over time. The remarkable discoveries by Ng et al. and others of ancient RNA viral genomes [26,27,28] provide hope in the pursuit of identifying more ancient viruses that shed light into the origins of contemporary viral mechanisms.

## Figures and Tables

**Figure 1 viruses-13-00493-f001:**
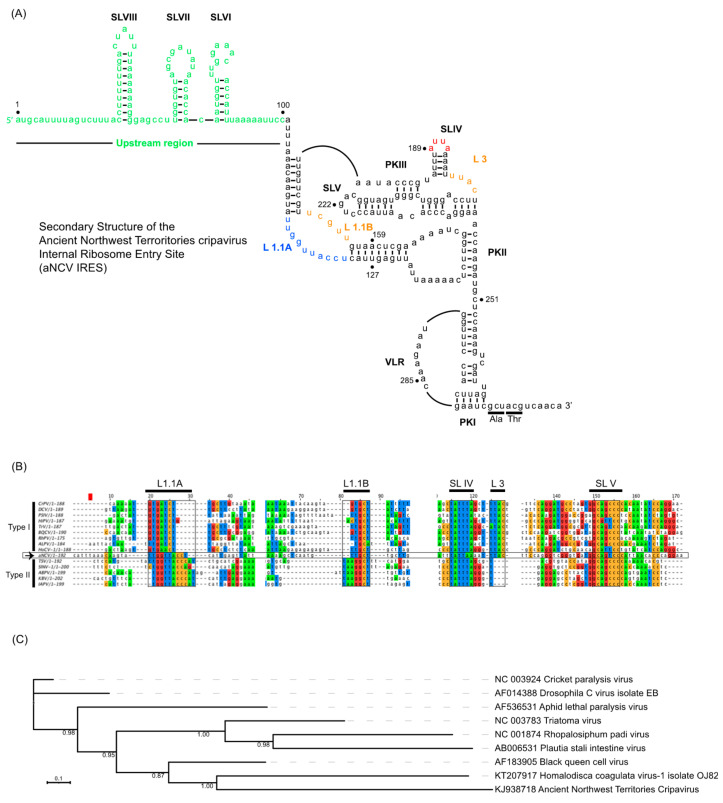
(**A**) Secondary structure model of the ancient Northwest Territories Cripavirus (aNCV) intergenic region (IGR) (accession KJ938718.1) Pseudoknots (PK) I, II, III are indicated. Orange-colored nucleotides at Loop (L)1.1B, L3 denote sequences conserved in Type I IGR internal ribosome entry site (IRES), and blue-colored nucleotides at L1.1A indicate sequences conserved in Type II IGR IRES. Red-colored nucleotides at stem loop (SL) IV are unique in sequence in aNCV IGR IRES. Green-colored nucleotides represent IGR sequences outside the core IRES. The predicted start codon is GCU adjacent to the PKI domain. Numbering refers to the nucleotide position in the IGR as the genome of aNCV is not complete. (**B**) Alignment of Type I and II IGR IRESs with aNCV IGR. (**C**) Maximum-likelihood phylogeny of the *Dicistroviridae* Type I IRES nucleotide sequences. SH-like branch support values are indicated at nodes and the maximum-likelihood scale bar indicates average residues substitution per site.

**Figure 2 viruses-13-00493-f002:**
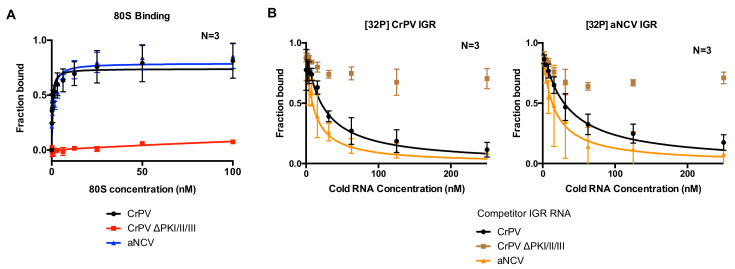
Affinity of 80S-aNCV IGR IRES complexes. (**A**) Filter binding assays. [^32^P]-aNCV IGR IRES, cricket paralysis virus (CrPV) IGR IRES or mutant (ΔPKI/II/III) CrPV IGR IRES (0.5 nM) were incubated with increasing amounts of purified salt-washed 80S. The fractions bound were quantified by phosphor-imager analysis. (**B**) Competition assays. Quantification of radiolabeled 80S-CrPV IGR IRES (left) or 80S-aNCV IGR IRES (right) complex formation with increasing amounts of cold competitor RNAs (aNCV IGR IRES, CrPV IGR IRES or mutant (ΔPKI/II/III) CrPV IGR IRES). Shown are the averages ± standard deviation from at least three independent experiments.

**Figure 3 viruses-13-00493-f003:**
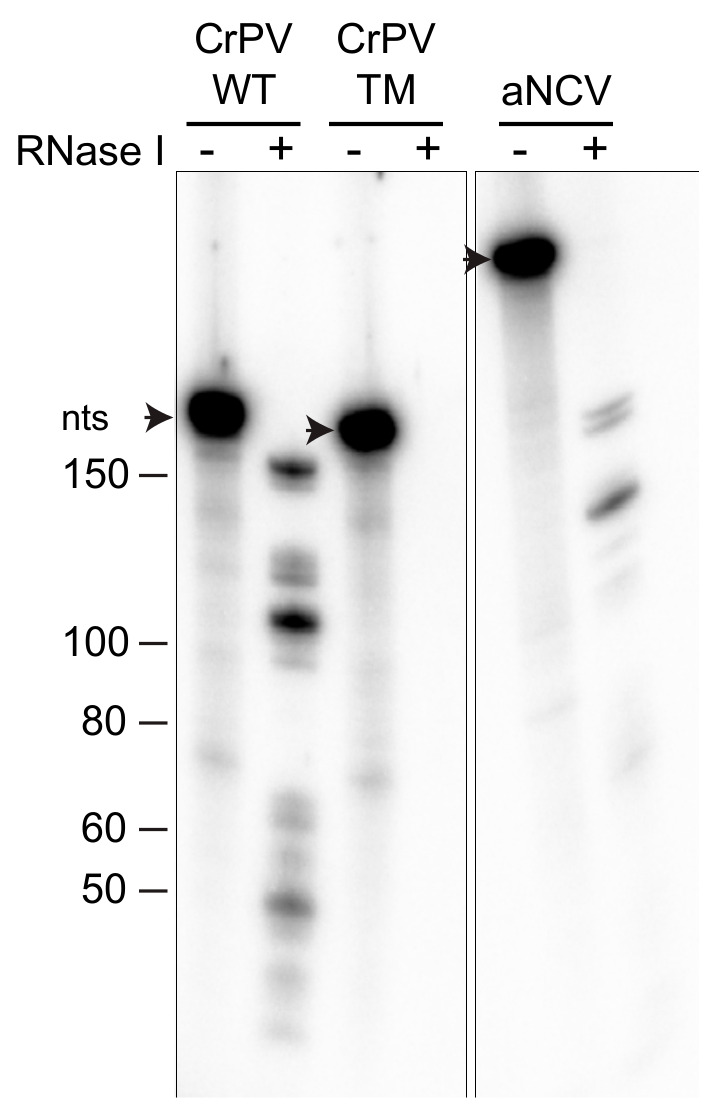
aNCV IGR IRES bound to 80S is RNase I-resistant. Radiolabeled RNAs were incubated with 80S (0.6 μM) for 20 min before adding RNase I (1 U) for 1 hr. RNAs from RNase I treated/untreated reactions were TRIZOL-extracted and loaded onto urea-PAGE and visualized by phosphor-imager analysis. Shown is a representative gel from at least two independent experiments.

**Figure 4 viruses-13-00493-f004:**
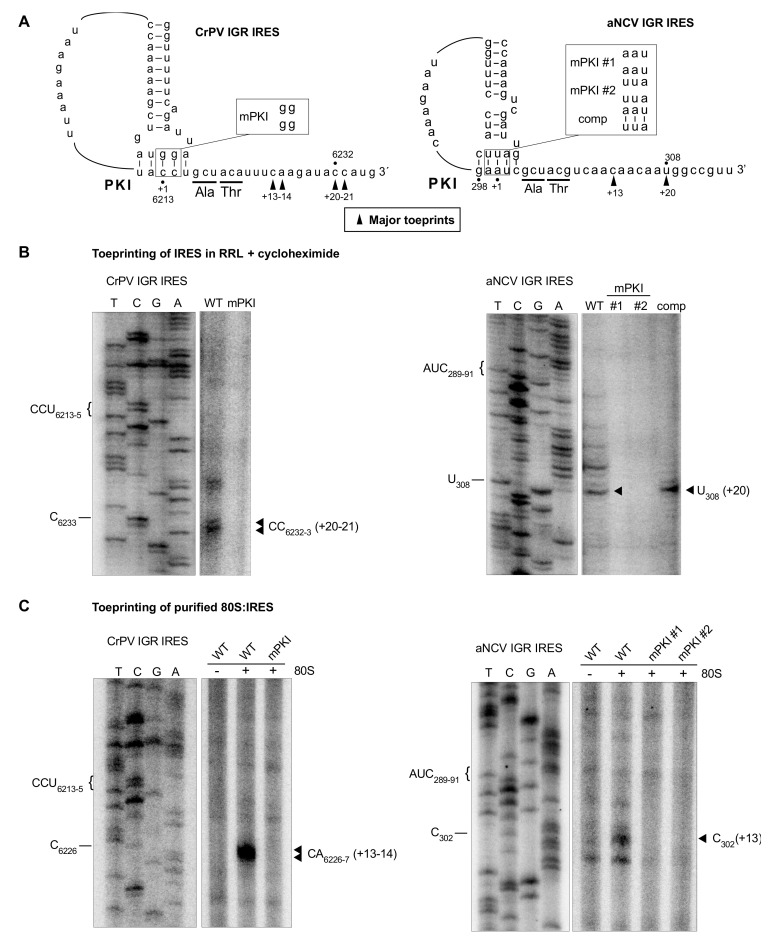
Toeprinting analysis of 80S-aNCV IRES complexes. (**A**) Schematic of IGR IRES PKI region. Mutations within the PKI region are annotated. (**B**) Primer extension analysis was performed on in vitro transcribed bicistronic RNAs containing the indicated wild-type or mutant CrPV IRES (left) or aNCV IRES or CrPV IRES (right) incubated in rabbit reticulocyte lysate (RRL) pretreated with cycloheximide (0.68 mg/mL). For the aNCV IRES, the sequencing ladder corresponds to nucleotides 279–331. The major toeprint is indicated at right, which is twenty nucleotides downstream of AUC_289-91_ of PKI given that the A is +1. (**C**) Primer extension analysis on in vitro transcribed bicistronic RNAs containing the indicated wild-type or mutant aNCV IRES incubated with purified ribosomes. Sequencing ladder corresponds to aNCV IRES nucleotides 269–319. The major toeprint is indicated at right, which is 13 nucleotides downstream of AUC_289-91_. Shown is a representative gel from at least three independent experiments.

**Figure 5 viruses-13-00493-f005:**
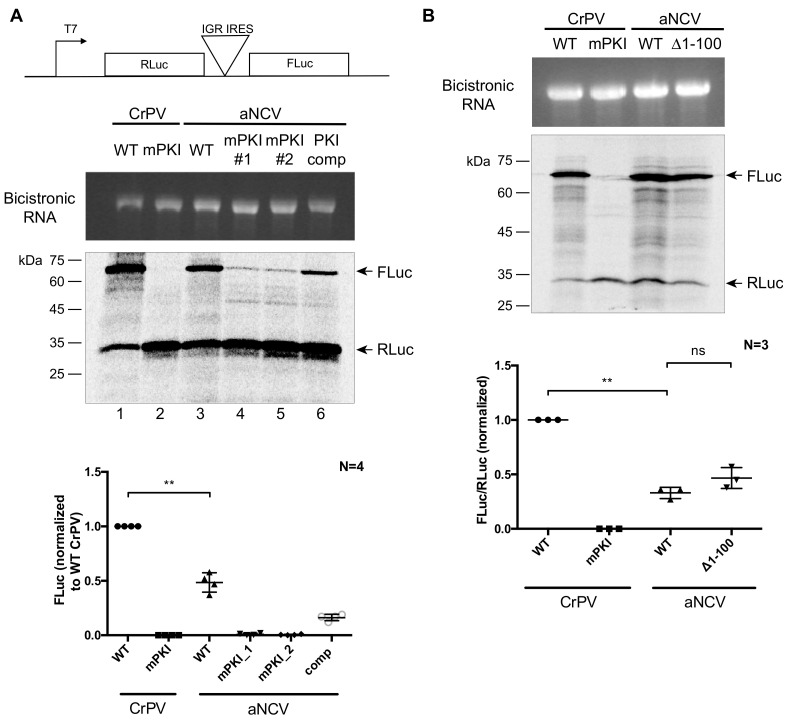
In vitro translation assays in RRL. (**A**) Schematic of bicistronic reporter construct containing the IRES within the intergenic region. (**B**) In vitro-transcribed bicistronic reporter RNAs were incubated in rabbit reticulocyte lysate (RRL) for 60 min in the presence of [^35^S]-methionine/cysteine. The reactions were loaded on an SDS-PAGE gel, which was then dried and imaged by phosphor-imager analysis. (top) Integrity of the in vitro transcribed RNAs are shown. (bottom) Quantification of the radiolabeled Renilla (RLuc) and firefly (FLuc) luciferase proteins. ns, *p* > 0.05; **, *p* < 0.005. Shown are the averages ± standard deviation from at least three independent experiments.

**Figure 6 viruses-13-00493-f006:**
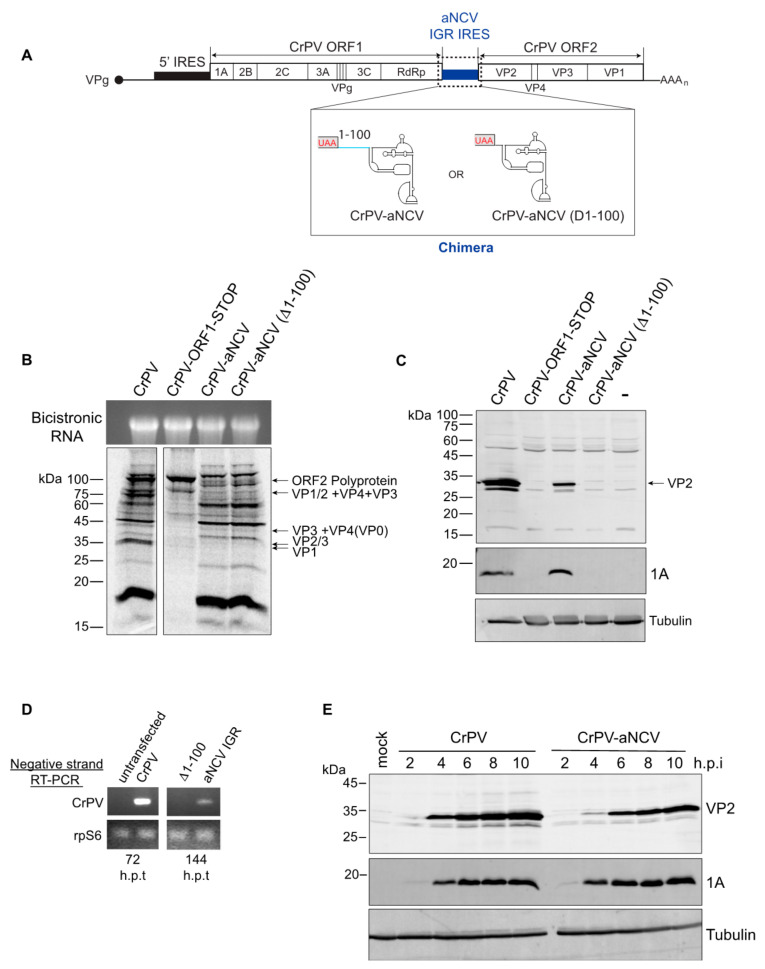
Chimeric CrPV-aNCV clone is infectious in *Drosophila* S2 cells. (**A**) Schematic of the chimeric CrPV-aNCV clone replacing CrPV IGR IRES with that of the aNCV IRES. (**B**) In vitro translation of CrPV and CrPV-aNCV RNAs in Sf21 extracts. CrPV-ORF1-STOP contains a stop codon within the N-terminal ORF1, thus preventing expression of the non-structural proteins. Reactions were resolved by SDS-PAGE and visualized by autoradiography. (**C**) Immunoblotting of CrPV VP2 structural protein and 1A non-structural protein. (**D**) RT-PCR of viral negative-strand RNA from Drosophila S2 cells transfected with the indicated viral clone RNAs at 72 and 144 h after transfection. (**E**) Immunoblotting of CrPV 1A and VP2 proteins from lysates of Drosophila S2 cells infected with CrPV or CrPV-aNCV chimera (MOI 5) at the indicated h.p.i. Shown are representative gels from at least three independent experiments.

## Data Availability

Not applicable.

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
