# Peer review of "Resurrection of a Viral Internal Ribosome Entry Site from a 700 Year Old Ancient Northwest Territories Cripavirus"

_viruses, 2021, doi:10.3390/v13030493_

Round 1

Reviewer 1 Report

Dicistrovirus intergenic region (IGR) IRESs mediate initiation by a remarkably streamlined mechanism that involves their binding directly to the ribosome without the involvement of initiation factors, initiator tRNA or an AUG initiation codon. This mechanism has no parallel to previously described mechanisms of translation initiation, leading to interest in its evolutionary origins. In the present report, Wang et al. have characterized an IGR IRES identified by metagenomic methods in a partial dicistrovirus genome ("ancient Northwest territories cripavirus" (aNCV)) recovered from ~700 years old frozen animal feces. They determined that this IGR contains an IRES that has a mixture of sequence and structural characteristics of the two currently recognized major classes of dicistrovirus IRES, that it bound directly to mammalian ribosomes, supported initiation of translation in mammalian and insect cell-free extracts, and could replace the Cricket paralysis virus (CrPV) IGR IRES in a full-length CrPV clone to yield a stable infectious chimeric genome. Although these studies do not directly advance understanding of the evolutionary origins of IGR IRESs, they will be of interest because the aNCV IGR IRES is functional and has characteristics that appear to be an amalgam of type I and type II IGR IRESs, and because the authors' success in introducing a heterologous IGR IRES into an infectious CrPV clone and assaying its ability to support viral infection validates an important approach to studying the function of IGR IRESs from incomplete genomic fragments in the context of viral infection.

SPECIFIC COMMENTS

  1. The author's phylogenetic analysis of IGR IRESs (Fig. 1C) is interesting, but the method used (Section 2.13) is not fully described. The authors should state whether aligned sequences were trimmed prior to phylogenetic analysis, explain why they used the HKY85 +G substitution model and mention caveats regarding the use of DNA substitution models for structured RNAs that contains single-stranded elements.
  1. Sections 3.2, 3.4. The authors assayed binding of 80S ribosomes to the aNCV IGR IRES. Did they assay binding to 40S subunits alone? It would be of interest to include such data, because it is a characteristic that differentiates the recently described Halastavi arva virus IGR IRES (which binds only to 80S ribosomes) from conventional type I and type II IRESs (which bind to 40S subunits and to 80S ribosomes).  
  1. Line 50. The blanket statement that dicistrovirus genomes contain a type III IRES in their 5'UTR is certainly incorrect, because the R. padi virus IRES (Terenin et al (2005) Mol. Cell. Biol. 25:7879–7888) and the CrPV IRES are thought to use very different mechanisms for initiation. Thus, there are at least two types of dicistrovirus 5'UTR IRES. The designation of the CrPV IRES as "type III" is by no means definitive, because its structure does not resemble those of conventional HCV-like type III IRESs. The authors should be more circumspect in their statements.
  1. Line 330. The statement "the full-length CrPV IGR IRES (208 nucleotides)" is incorrect, because the IRES including the GCU initiation codon is only 191nt long (nt. 6029-6219).
  1. Line 70. References 12, 13 concern bacterial ribosomes and don't have anything to do with cryoEM analysis of binding of IGR IRESs to different sites on the ribosome.
  1. There are numerous typographical errors that should be corrected e.g.
  2. a) Lines 33-34. The acronym "eIFs'" derived from eukaryotic initiation factors, not eukaryotic translation
  3. b) Line 34. "5'cap scanning mechanism" should be "5'cap-dependent scanning mechanism".
  4. c) Line 35. "Scanning of" should be replaced by "Scanning by".
  5. e) Lines 80-81. What are "penaeid shrimp outbreaks"? The authors presumably mean outbreaks of disease in penaeid shrimp.
  6. f) Line 170. The greek letter 'Delta' appears not to be connected to any other element of text.
  7. g) Lines 291-292 " Homalodisca coagulate virus-1" should be "Homalodisca coagulata virus-1 ".
  8. h) Line 510. " pseudotranslocation ", not "pseduotranslocation ".

Author Response

Reviewer 1

Comment 1: The author's phylogenetic analysis of IGR IRESs (Fig. 1C) is interesting, but the method used (Section 2.13) is not fully described. The authors should state whether aligned sequences were trimmed prior to phylogenetic analysis, explain why they used the HKY85 +G substitution model and mention caveats regarding the use of DNA substitution models for structured RNAs that contains single-stranded elements.

RESPONSE: Thank you for the comment. The aligned IRES sequences were not trimmed and the method section is updated (page 6, line 271). The HKY85 +G substitution model was selected using standard model selection software, as mentioned in line 298 “Maximum-likelihood trees were constructed with PhyML 3.0 and the HKY85 +G nucleotide model was selected using Smart Model Selection”, as this is the best approach for selecting the best model considering what is available within the phylogenetic software. In general, the majority of substitution models are not optimal for analyzing viruses, especially RNA viruses but we selected the model that was best considering the possibilities available to us. CAVEATS: Functional constraints leading to highly conserved secondary structure are present more so in RNA molecules than DNA molecules. So compensatory mutations for the preservation of secondary structure is important in RNA evolution but less so in DNA evolution. This would not be accounted for in a DNA substitution model. While it isn’t ideal to use a DNA substitution model on an alignment of RNA molecules, we’ve tried to mitigate some of this bias by using structural information to inform and construct the alignment. However, it is very likely that considering the model used, the IRES sequences analyzed may appear more divergent than they truly are. The same biases would however apply to all the sequences and therefore the analysis can still be informative within its constraints.

Comment 2: Sections 3.2, 3.4. The authors assayed binding of 80S ribosomes to the aNCV IGR IRES. Did they assay binding to 40S subunits alone? It would be of interest to include such data, because it is a characteristic that differentiates the recently described Halastavi arva virus IGR IRES (which binds only to 80S ribosomes) from conventional type I and type II IRESs (which bind to 40S subunits and to 80S ribosomes).  

RESPONSE: Thank you for the comment. We agree that there are now distinctions in the ribosome assembly pathways on different IGR IRESs. However, given that the aNCV IGR IRES contains SLV and SLIV that closely resemble Type I IGR IRESs, it is likely that the aNCV IGR IRES will bind to 40S similarly. In the case of the Halastavi arva virus IGR IRES, it is clear that SLV and SLIV are missing and thus pointed to a distinction in how it binds to 40S/80S. Given the predicted structure of the aNCV IGR IRES, we focused on the end result of 80S binding but we agree with the reviewer that testing for 40S binding should be performed at a later time to confirm the 80S assembly pathway. 

Comment 3: The blanket statement that dicistrovirus genomes contain a type III IRES in their 5'UTR is certainly incorrect, because the R. padi virus IRES (Terenin et al (2005) Mol. Cell. Biol. 25:7879–7888) and the CrPV IRES are thought to use very different mechanisms for initiation. Thus, there are at least two types of dicistrovirus 5'UTR IRES. The designation of the CrPV IRES as "type III" is by no means definitive, because its structure does not resemble those of conventional HCV-like type III IRESs. The authors should be more circumspect in their statements. 

RESPONSE: We thank the reviewer for pointing out this overstatement. We have now changed the original sentence “Translation of the first ORF is directed by a type III-like IRES which is located at the 5’ UTR” to “Translation of the first ORF is directed by an IRES located at the 5’ UTR” (Page 2, line 52). Comment 4: The statement "the full-length CrPV IGR IRES (208 nucleotides)" is incorrect, because the IRES including the GCU initiation codon is only 191nt long (nt. 6029-6219). Response: Thanks for pointing out the confusion here. The CrPV IGR IRES used in the assay includes nucleotides upstream downstream of the IRES, which ensures the RNA to fold properly. However, the reviewer is correct that the core IRES itself is 188 nucleotides. We changed the original statement “…compared to the full-length CrPV IGR IRES (208 nucleotides)” to “…compared to the full-length CrPV IGR IRES (188 nucleotides)” (Page 7, line 341). Comment 5: References 12, 13 concern bacterial ribosomes and don't have anything to do with cryoEM analysis of binding of IGR IRESs to different sites on the ribosome. Response: Thank you for this. The corrected references “Spahn, C. M. T.; Jan, E.; Mulder, A.; Grassucci, R. A.; Sarnow, P.; Frank, J. Cryo-EM Visualization of a Viral Internal Ribosome Entry Site Bound to Human Ribosomes: The IRES Functions as an RNA-Based Translation Factor. Cell 2004, 118 (4), 465–475.” and “Abeyrathne, P. D.; Koh, C. S.; Grant, T.; Grigorieff, N.; Korostelev, A. A. Ensemble Cryo-EM Uncovers Inchworm-like Translocation of a Viral IRES through the Ribosome. Elife 2016, 5, e14874.” are added (Page 2, line 72). 

Comment 6: There are numerous typographical errors that should be corrected

RESPONSE: Thank you for careful editing. We have gone through the manuscript carefully and corrected all typographical errors. 

Reviewer 2 Report

The manuscript by Wang et al. describes ribosomal binding and translation initiated by aNCV IGR IRES. The results demonstrate that the aNCV IGR IRES has unique and conserved features, efficiently binds ribosomes, and can be used to initiate translation of proteins, including the rescue of an infectious clone. The rationale for experiments is well-explained, and the results are clear. The discussion is concise, perhaps reflecting the truncated sequence available for aNCV, but highlights the evolutionary importance of characterizing the IRES of an ancient RNA virus. Adding to the interest is the utility of IRES sequences in genetically modified systems. Minor comments are below.

1) Do experiments described in 3.4/Fig 4 (specifically lines 356-361) prove that GCU is the starting codon or that disrupting the predicted structure is detrimental to activity?

2) How come the aNCV (D1-100) was not used for the experiments demonstrating ribosome binding?

Author Response

Reviewer 2 

Comment 1: Do experiments described in 3.4/Fig 4 (specifically lines 356-361) prove that GCU is the starting codon or that disrupting the predicted structure is detrimental to activity? 

RESPONSE: We thank the reviewer for this error. We have now changed the sentence “thus supporting the conclusion that ribosomes initially assembled on the aNCV IGR contains the AUC codon in the ribosomal A site and the adjacent GCU codon is the initiating start site.” to “….thus supporting the conclusion that ribosomes initially assembled on the aNCV IGR contains the AUC codon in the ribosomal A site and subsequently, after the first pseudotranslocation step, the adjacent start site GCU codon occupies the A site where IRES initiation starts from”. This statement now better reflects the results (Page 8, line 381-383). 

Comment 2: How come the aNCV (D1-100) was not used for the experiments demonstrating ribosome binding? 

RESPONSE: We thank the reviewer for this comment. Since the in vitro translation data shows that IRES activity is similar between the full-length and mutant aNCV (D1-99) IGR (section 3.5, 3.6), we did not pursue ribosome binding experiments as it was unlikely that the 1-99 sequence would affect ribosome binding. 
